# Analysis of Health in Environmental Assessments—A Literature Review and Survey with a Focus on Denmark

**DOI:** 10.3390/ijerph16224570

**Published:** 2019-11-18

**Authors:** Sarah Humboldt-Dachroeden, Birgitte Fischer-Bonde, Gabriel Gulis

**Affiliations:** 1Department of Social Science and Business, Roskilde University, Universitetsvej 1, 4000 Roskilde, Denmark; 2Fischer-Bonde Consulting, Asger Rygs Gade 18, 2.tv, 1727 København V, Denmark; birgittefischerbonde@gmail.com; 3Unit for Health Promotion Research, University of Southern Denmark, Niels Bohrsvej 9-10, 6700 Esbjerg, Denmark; ggulis@health.sdu.dk

**Keywords:** environmental impact assessment, strategic environmental assessment, health assessment, comparative analysis, policy review

## Abstract

In the European Union, the Environmental Impact Assessment (EIA) Directive (2014/52/EU) and Strategic Environmental Assessment (SEA) Directive (2011/92/EU) emphasise the assessment of population and human health. The directives require health to be considered within Environmental Impact Assessment (EIA) and Strategic Environmental Assessment (SEA). To date, health is mainly considered in connection with negative environmental factors and in terms of risk assessments. The integration of health in EIA as well as SEA has not been investigated in a Danish context, and this study aims to address the missing knowledge. There is a need for a more comprehensive health assessment within EIA and SEA to comply with the EIA and SEA directives. An integration of health into EIA and SEA will ensure a sound examination of health determinants which can improve decision making and thus comprehensively promote and protect health. To establish the status of the inclusion of the assessment of impacts on health into EIA and SEA, a literature review was performed. In addition, a survey addressed to researchers and practitioners was conducted and analysed through a comparative analysis. The survey examined the needs of practitioners and researchers, focusing on the Danish context, regarding the inclusion of health into EIA and SEA. Enhanced intersectoral cooperation of the health and environmental sectors, more specific guidance documents, and underlying this, stronger political support, were identified among needs for more comprehensive health assessments.

## 1. Introduction

Environmental degradation, the loss of biodiversity and various consequences of climate change are threatening the planet and consequently also human health [1]. At the same time, the state of health is changing as shown by the Global Burden of Disease (GBD) studies [2]. This data can be used to examine a variety of health determinants. Health burdens revealed through the GBD study include the rise in non-communicable diseases (NCDs) such as cardiovascular diseases and diabetes [3]. Hence, assessing and considering health implications becomes increasingly important to avert further negative implications to population and human health. To account for the environment and its preservation, the European Commission (EC) set out directives to mitigate those effects. Environmental assessments such as the Environmental Impact Assessment (EIA) and Strategic Environmental Assessment (SEA) were implemented via directives making them mandatory within the European Union (EU). EIA and SEA are both tools to assess the environmental impacts of plans, programs and projects. They ensure that possible implications on the environment, negative and positive, are being revealed to support the decision-making process. Among environmental considerations, health is being assessed, but only to a certain extent. The amended version of the 2011 EIA Directive (2011/92/EU), the EIA Directive (2014/52/EU), specifically notes population and human health as a category for assessment compared to the previous Directive (2011/92/EU), which only mentioned human beings [4,5]. The SEA Directive (2001/42/EC) also emphasises a protection of human health. Since the directives have entered into force, the coverage of health impacts is often limited [6]. EIA/SEAs mainly focus on negative environmental factors, such as adverse health effects resulting from noise, air quality or temperature. However, to comply with the directives, an emphasis must be put on considering population and human health beyond environmental health impacts. This can foster a broader assessment of determinants which can lead to the promotion and the protection of health, especially considering NCD diseases in the EU [7,8].

The World Health Organisation (WHO) has defined health as a “*state of complete physical, mental and social well-being and not merely the absence of disease or infirmity*” [9]. With this definition, the WHO made a first approach to encompass all aspects of health. The WHO is supporting the integration of a broad health assessment within EIA/SEAs [7,8]. A model often referred to, and influenced by the broad definition of health, is the *Determinants of Health Model* by Dahlgren and Whitehead [10], which was further developed by Barton and Grant [11]. It shows that health can be affected by a variety of determinants. The influences they describe are levelled upon each other, considering age as well as genetic factors, starting with the closest factors influencing a person, the individual lifestyle factors, then the social and community networks followed by general socio-economic, cultural and environmental factors [11].

The Health Impact Assessment (HIA) is a comprehensive and established tool to assess health, which could aid to elevate health assessments within EIA/SEAs with its methodologies and approaches [7]. An integration of a HIA or an extended health assessment into EIA/SEA could ensure a sound examination of health determinants, which can improve decision making of plans, programs and projects and thus promote as well as protect health.

There is no common approach to include health into EIA/SEA. The WHO has suggested that guidance should be developed. However, developing guidance documents has proved a challenge, since EIA/SEAs are comprehensive undertakings and can be conducted in many different areas [8]. In Annex I and II of the EIA Directive, there are approximately ten areas in which EIA/SEA must be conducted [4,12]. A guidance document that addresses health in environmental assessments must be broad enough to be applicable in diverse disciplines but specific enough to be ready to use. Consequently, this would lead to a variety of different outlines for guidance documents. These complex underlying structures challenge researchers and practitioners to establish comprehensive and mutually agreed guidance documents [8].

All EU countries have transposed the EIA and SEA directives differently and in their own languages. Due to the varying transpositions, the needs of practitioners naturally differ between the countries. Denmark was chosen to exemplify the transposition of the EC directives and to assess the needs of Danish practitioners.

In Denmark, the EIA Directive was transposed into the law on environmental assessments of plans, programs and concrete projects (Danish: *Bekendtgørelse af lov om miljøvurdering af planer og programmer og af konkrete projekter* (VVM)), hereafter named the VVM law [13]. The VVM law comprises the formerly separated laws on the environmental assessment of plans and programs with the environmental assessment of concrete projects. That means that the legislation of the EIA and SEA Directive were merged and became one law [14]. The VVM law does not define population and human health.

The Danish Environmental Protection Agency, operating under the Ministry of Environment and Food of Denmark, are in the process of developing a guidance document. However, difficulties are the coherent translation of terms such as population health and public health into Danish as well as a lack of definition thereof [14].

Aiming to understand barriers and facilitators of health inclusion into EIA/SEA, the focus of this study was set on Denmark, as there is a gap in the literature concerning this issue. This article aims to aid revealing different perspectives of researchers and practitioners, with a focus on Danish practitioners, regarding the inclusion of health into EIA and SEA. In the context of EIA/SEA, practitioners are the workforce that operate within various branches of the industry to conduct EIA/SEA upon request of the industry or of the authorities, including municipalities. On the other hand, EIA/SEA researchers are those who produce scientific and theoretical background on EIA/SEA implementation.

## 2. Materials and Methods

### 2.1. Literature Review

The databases PubMed and ScienceDirect were searched for scientific literature, and no quality assessment was performed. The literature search included relevant articles in English from 2014, the year in which the amended EIA Directive was published, until February 2019. A modified search string was used which included the terms “Environmental Impact Assessment”, Strategic Environmental Assessment”, Health Impact Assessment”, and also “project”, “programme”, “plan” and “policy”. Included were articles that mentioned EIA and SEA which incorporated a HIA or health assessment. EIA or SEA policies, plans, programmes and projects as well as an elaboration of the institutionalisation, integration or implementation of EIA, SEA and HIA was reviewed. The articles were selected based on inclusion and exclusion criteria—see Table 1—and analysed based on their scope of health determinants. Table 2 lists the scope, which was drawn from Nowacki [7], who based it on the Determinants of Health Model by Barton and Grant [11]. The focus was on extracting knowledge on how and to what extent health was included in environmental assessments.

As no articles were found pertaining to Danish EIA/SEA implementation, a grey literature search was conducted to locate EIA/SEA reports. The grey literature was located through a web search via google. Danish EIA reports were found on the website of Danish authorities that conduct EIAs. The grey literature was analysed based on their scope of health determinants—see Table 2.

### 2.2. Survey

The survey was conducted in 2019, from March to April. The aim of the survey was to reveal the needs of practitioners and researchers to integrate health into EIA/SEA. Target populations were researchers within the EU who conduct research on impact assessments such as EIA, SEA and HIA. Practitioners who implement EIA, SEA and HIA from the EU and especially from Denmark were also targeted.

The questionnaire produced both quantitative and qualitative data through closed- and open-ended questions. The standard language of the questionnaire was English. Due to the focus on Danish practitioners, the overall introduction, sub-introductions and definitions were translated into Danish.

Prior to initiating the survey, a small pilot study was conducted. Danish researchers, public health master students and individuals not connected to the topic of EIA/SEA and HIA tested the survey regarding coherence, language, comprehensibility and relevance to the subject.

The questionnaire was anonymous and no sensitive or personal information such as ethnicity, gender, religion or sexual orientation was gathered.

Three different methods were employed to gather respondents: The survey link was distributed via email to participants of the 2018 one-day workshop “Miljøvurderingsdag” (Environmental Assessment Day) in Denmark. Participants of the workshop were EIA and SEA practitioners and researchers who discussed challenges and opportunities within EIA.The link to the survey was shared through the LinkedIn group “HIA—Health Impact Assessment Group”. Members of this group consist of practitioners and researchers in the field of HIA.The link to the survey was distributed to all participants of the “Human Health in Environmental Impact Assessment” conference, which took place in March 2019 at the WHO European Centre for Environment and Health office in Germany.

#### Comparative Analysis

A comparative analysis of the survey results was conducted. This type of comparative analysis is a case-based method, in which there are few groups, but many variables. The groups were the researchers and practitioners and the variables were the questions asked in the survey. Respondents were categorised as researchers and practitioners based on their self-categorisation within the survey. Practitioners conduct EIA/SEA upon request of the industry or governments. Researchers are those who produce scientific as well as theoretical background on EIA/SEA utilisation and implementation. Another feature of the comparative method is an understanding of the different characteristics of the groups. The variables assessed can have different functions or impacts on the respective group. Due to the need for clearly defined groups in comparative studies, the results are rarely generalisable and transferable to other groups or contexts [15].

Most of the questions were identical for both researchers and practitioners. However, some questions were tailored to the context of the two groups. These adjustments mainly resulted out of the difference in work of researchers, who conduct research and of practitioners, who conduct EIA/SEA or HIA. Those questions were only altered to the extent that they could still be compared to each other. Of the 34 questions, seven were correspondingly tailored to the researcher’s context and to the practitioner’s context. The remaining 27 questions were identical, see Appendix B (Table A1) for the full questionnaire.

## 3. Results

### 3.1. Literature Search

Besides background material, the literature search revealed case studies which shed light on methodological approaches, health inclusion strategies and other political or social issues that played a role in the assessment. The search identified 2167 scientific articles—of which, 21 case studies fit the inclusion criteria. See Appendix A (Figure A1) for the literature screening process. These case studies showed a clear inclusion of health or HIA into EIA/SEA. Relevant articles were selected to exemplify three European cases. Two Danish cases were identified through a grey literature search—see Table 3. Those cases were selected, as they showed the current practice of health assessments in EIA/SEA within Europe and in Denmark. EIA and SEA are usually written in the form of a report and are mainly the language of the origin country. Hence, it was chosen to investigate European EIA/SEAs through scientific articles written in English to circumvent this issue. This gave the opportunity to examine European examples of environmental assessments. However, the literature review is not comprehensive, since only a few EIA/SEA reports are transformed into scientific articles, and because scientific articles of EIA/SEAs often only describe the process of the environmental assessments or specific aspects of them. To analyse the adoption of EIA/SEA within a country, Denmark was selected. Thereby, an examination of the legislatives resulting from the EIA and SEA directives and of EIA reports was conducted.

#### 3.1.1. European Environmental Assessments

The SEA conducted on an urban development plan for the expansion of the Tivat airport in Montenegro was described as an inclusive participatory approach. Within the SEA, the scope comprised mainly environmental determinants, but also referred to identified vulnerable facilities, such as educational and medical institutions. The extension of the airport was related to economic benefits for the region, especially due to an increase in tourism in the area. Josimović et al. described the SEA as encompassing many environmental and socio-economic factors, but the focus of the article was mainly on the environmental factor of noise [16]. Although mainly noise and negative health outcomes were described, there was an inclusive approach regarding health determinants, since impacts on vulnerable facilities were considered.

Carvalho et al. investigated the implementation of the SEA on the railway line at the Lithuanian and Latvian border. The article examined the approach of the SEA and compared it to two other assessments. In addition to the article, an official information document for the convention of the SEA and a dissertation on the topic were identified as grey literature. The documents provide insight into the assessment of biophysical, social and economic determinants. Health was not assessed separately but included into social determinants. The public were engaged in all steps of the SEA [17].

In Serbia, a regional waste management plan was proposed for the Kolubara region which covered eleven municipalities. A SEA was conducted on the waste management plan, which was documented by a scientific article, examining the method of the SEA. A chapter in the book “Environmental Management” by Sarkar further described the implementation of SEA in Serbia, exemplifying the waste management case [18,19]. The SEA identified negative as well as positive health-related impacts and established channels for public participation. The scope mentioned “receptors” of the environment, landscape and population, human health and socio-economic development. Regarding the health and socio-economic determinants, specific objectives were implemented to foster economic growth through creating jobs and by protecting human health from emissions [18].

#### 3.1.2. Danish Environmental Assessments

To analyse the inclusion of health determinants in the Danish EIA practice, two EIAs were selected: a railway workshop in Fredericia by the Danish State Railways (DSB), and a biogas plant at a farm [20,21].

DSB are planning the purchase of new electric trains, for which a new maintenance workshop is needed. The EIA of the maintenance workshop considered mainly environmental factors such as noise and vibrations resulting from increased traffic during the construction phase. It further assessed the risks to human health and the environment regarding demolishing of the current maintenance workshop building, which can possibly create harmful waste. The environmental factors were concluded to have either none, small or insignificant impacts on the environment or the population and human health [19].

Factors related to population and human health covered in the EIA of the biogas plant by the municipality of Svendborg mainly covered environmental factors throughout the construction and operational phases. Noise, vibrations and safety issues related to increased traffic and production were analysed. It was concluded that the factors would not have a significant effect on population and human health. Production-related smell and infection risk from manure used to produce biogas, were minimal and within the limit values. In turn, the production of biogas can have health benefits for the affected population due to a decrease in bacteria in the fertiliser after the process of extracting the gas. Furthermore, the area is used for general farming activities, including heavy duty traffic. The traffic from the biogas plant would not result in a substantial increase in traffic. Due to the above-mentioned factors and the low number of residents in the area, none of the assessed factors were found to have significant negative impacts on human health [20].

### 3.2. Survey

In total, 42 respondents completed the survey. Table 4 summarizes the basic characteristics of the respondents. Although the sample is rather small, it was considered appropriate for the context of this study. Considering the size of Denmark and possible work opportunities in the area of conducting EIA, it was presumed that a solid number of practitioners was reached. The remaining seven practitioners who came from EU countries not including Denmark, supported the results in the EU context. Regarding researchers, the sample size was smaller. Opinions that were collected might not represent the opinion of all researchers in the EU who conduct research on the topic of EIA/SEA, HIA or health in EIA/SEA. Hence, results were treated as the views of a non-representative group of researchers. All in all, responses of participants were interpreted as a direction and orientation, regarding their understanding and needs of health in EIA/SEA.

Comparing Danish practitioners to all practitioners, most answers provided similar or the same results. To illustrate variations pertaining to the Danish context, Danish practitioners are extracted from the group of all practitioners and are presented separately. With 66%, the majority of practitioners were employed in regional or local environmental authorities and in private companies as environmental consultants, and 34% were employed in national health and environmental authorities, health or environmental inspectorates, private companies as health consultants or in other private companies. Researchers covered a broad range of occupations including international health organisations, regional or local health and environmental authorities, public health and environmental institutes, academic units and non-governmental organisations.

While conducting EIA/SEAs, 20 of all the 32 practitioners never consulted with public health or health professionals. Only 14 of all practitioners—of which, 10 were Danish practitioners—consulted with health professionals. Three of all researchers who participated in the survey assisted in the process of conducting EIA/SEA.

Practitioners were asked whether they had ever integrated health into EIA/SEA, and researchers were asked whether they had ever conducted research on the integration of health into EIA/SEA. 66% of practitioners had integrated health into EIA/SEA and 80% of researchers had conducted research on the integration of health into EIA/SEA. Respondents’ work experience was then compared to their preferences regarding the inclusion of HIA into EIA/SEA. In this question, their preference was assessed only regarding an inclusion of HIA and not regarding a health assessment. While practitioners and researchers both had experience integrating health into EIA/SEA, integrating a stand-alone HIA was less preferred by the respondents. In total, 50% of all practitioners—of which, 44% were Danish practitioners—and 40% of researchers answered that HIA should be conducted within EIA/SEA. A Danish EIA practitioner stated that HIA should not be integrated into EIA, but rather be an independently required assessment if the screening process revealed the need for it. Respondents who stated that HIA should be conducted within EIA argued with the importance of health, and that *“people are part of the environment”* or suggested that health considerations should be mainstreamed within impact assessments.

Since the majority of practitioners and researchers had integrated health into EIA/SEA, a more detailed understanding of perceived barriers and facilitators for the inclusion of health into EIA/SEA was investigated. Only lack of experiences/competencies within the field was in the top three barriers for both practitioners and researchers. For practitioners, it represented the main barrier for the inclusion of health into EIA/SEA (50%). The main barrier for researchers was institutional barriers between different sectors (60%). Table 5 lists the three main barriers of practitioners and researchers to include health into EIA/SEA.

On the other hand, the three main facilitators to include health into EIA/SEA provided more overlaps. Guidance documents, collaboration between different sectors and political support were all listed as main facilitators, as shown in Table 5. The main facilitator in the researchers’ opinion was collaboration between different sectors. This was in coherence with the main barrier stated to be institutional barriers between different sectors. Further, practitioners stated that the lack of guidance documents was a barrier and the availability of guidance documents an important facilitator to include health into EIA/SEA.

Different sources can be utilised by practitioners to conduct EIA/SEAs. In the survey, options for different types of sources encompassed scientific literature, guidance documents, expert opinion, grey literature, policy documents and statistical databases. Questions were tailored to the context of practitioners and researchers. Practitioners were asked, which sources they use while conducting EIA/SEA and researchers were asked, which sources they think should be used while conducting EIA/SEA. In the opinion of researchers, all sources were important. Statistical databases, expert opinions and scientific literature all obtained 80% response as the most important sources. The three most important sources according to all practitioners were guidance documents, scientific literature and expert opinions. Guidance documents with 91% response displayed the most important source for practitioners—see Table 5.

Among the practitioners, there was consensus regarding the need for more guidance documents generally, specific to the type of documentation (plan, programme, project, policy) and to the area of EIA/SEA. Most researchers also saw a need for more guidance documents generally and specific to the type of documentation. Of the ten researchers, only four responded that more area specific guidance material is needed. However, five researchers stated that they were not sure whether there is a need for guidance documents to better distinguish the different areas.

Since the EIA and SEA directives mention population and human health, the survey investigated the perceived requirements of competencies to assess health. Competencies to choose from in the survey were medical background, public health background, health-related training, health-related workshop and no health-related competencies. With 69% of practitioners and all ten researchers, both sides agreed that a public health background should be required to assess population and human health within EIA/SEA. This was followed by health-related training by 50% of practitioners and 70% of researchers. Practitioners favoured neither a medical background nor no health-related competencies. In contrast, a medical background was emphasised by researchers.

## 4. Discussion

### 4.1. Integrating Health into Environmental Assessments

When asked about the HIA, only a few Danish practitioners knew what it was. This may be on account that HIA is neither mandatory nor a commonly used assessment in Denmark. Research into the use of HIAs in Denmark is outdated and official Danish guidelines are more than ten years old [22]. These might be reasons why some Danish practitioners dismissed the idea of conducting HIA within EIA. Further, several Danish practitioners opposed integrating other health-related assessments within EIA/SEA, since they viewed them as already very extensive reports. Other Danish practitioners felt confident that health was already sufficiently considered in the assessments.

Another contributing factor to the opposition towards HIA as a stand-alone document could be the current nine to twelve months approval process for EIAs in Denmark [14]. The development of an additional separate assessment would presumably prolong the process. Nevertheless, most respondents of the survey, also Danish practitioners, either believed that HIA should be conducted within EIA/SEA or that HIA should be conducted depending on the context of the specific case. Practitioners and researchers mainly felt that combining HIA with EIA/SEA should depend on the context. This could indicate the respondents’ awareness of the SEA Directive and the amended EIA Directive to “*identify, describe and assess […] population and human health*” [4,12].

There are two common ways to integrate a comprehensive health assessment into EIA/SEA. One is to assess health impacts more comprehensively within the usual assessment of impacts. The other is to integrate a stand-alone HIA into EIA/SEAs. A third approach which could be further reviewed was brought forward by a practitioner. It was suggested to conduct a screening on health impacts with a subsequent decision on whether to conduct a full HIA.

Practitioners generally supported the integration of a more comprehensive health assessment but were simultaneously restrained. The lack of health-related training and the lack of guidelines for health in EIA/SEA posed barriers to include a comprehensive health assessment into EIA/SEA. This showed the practitioner’s understanding of environmental influences on health. Although some practitioners might acknowledge the interrelation of health and environment, most practitioners emphasised that the health assessment should pertain to environmental impacts such as “*soil, air, water, chemicals, noise, safety etc.*”, as expressed by a practitioner in the survey. Further, one third of all practitioners had never integrated health into EIA/SEA. This might have contributed to the lack of progress regarding practitioners’ experiences and competencies to include health into environmental assessments. The examination of the literature review pertaining to most EIA/SEAs also showed that impacts assessed were mainly related to environmental factors. This is in line with the WHO’s conception that health assessments within EIA/SEA are often constrained to environmental or biophysical factors [8]. This practice neglects a full range of determinants such as influences on health through lifestyle factors, social networks, socio-economic, cultural and environmental conditions. Integrating the *Determinants of Health Model* and assessing the scope in the light of the model could facilitate a more comprehensive health assessment. It could aid practitioners to detect interrelations between determinants and could thus enhance examining all important factors.

### 4.2. Engagement of Health Professionals

Within the survey, respondents emphasised that collaboration between the different sectors could facilitate the inclusion of health into EIA/SEA, whereas institutional barriers impede the inclusion of health into environmental assessments. However, practitioners raised concerns that a health assessment within EIA/SEAs could lead to a reduced environmental assessment. Accordingly, it was mentioned that health assessments should be conducted within other types of impact assessments, which focus more on social factors. Thus, it must be ensured that the environmental assessment itself will not be neglected through the integration of health. However, this issue can be circumvented by enhanced cross-sectional cooperation while conducting the assessment. Cooperation can facilitate communication between the sectors, which can lay the ground for a comprehensive assessment of both, the environment and health. Collaborating with health professionals could lead to training programs for practitioners to enhance health-related competencies within environmental assessments.

Respondents agreed that the lack of professionals, experienced or educated in health-related fields was a crucial barrier to include health into EIA/SEA. At the same time, practitioners and especially Danish practitioners, had little consultation with health professionals in the process of conducting EIA/SEAs, which might be a result of institutional barriers between the health and environment sectors. This also became apparent when investigating the SEAs of the Tivat airport extension in Montenegro, the waste management plan in Serbia and the high-speed railway in Lithuania [16,17,18]. The articles pertaining to the cases in Montenegro and Serbia provided no information on the involvement of the health sector in the process of the SEA [16,17,18].

The SEA in Montenegro focussed on negative impacts through the environmental factor noise, but vulnerable facilities such as health and educational facilities were mentioned. The engagement of stakeholders facilitated the consideration of some health determinants other than environmental factors [16]. Presumably, the engagement of health professionals could have contributed to a more comprehensive consideration of health impacts.

The SEA conducted on the waste management plan in Serbia encompassed environmental factors, built environment and housing as well as employment and livelihood. For human health, objectives addressed risk minimisation through emissions, health promotion through minimising “environmental problems” and the creation of jobs [18]. A more thorough investigation, especially concerning social and health impacts could have provided an opportunity to protect health of the population in the area. The consultation with health experts could have contributed to a more thorough assessment of potential health impacts.

In the case of Lithuania, the Ministry of Health was engaged within the SEA [23,24]. The scope of the SEA that was conducted in Lithuania investigated environmental factors but also considered social factors and thus also health impacts. Furthermore, health impacts specific to mental health were listed, as well as positive impacts, such as the impact of tourism on social and economic issues in remote regions [24]. The SEA showed an approach to assess impacts beyond environmental factors. The engagement of the Ministry of Health probably contributed to the extent of the examined impacts.

In Denmark, the survey indicated that cooperation between the health and the environmental sector while conducting EIAs was also limited. The VVM law in Denmark has only been in operation since 2017. Since then, population and human health must also be considered within Danish EIAs. As the results of the survey showed, the collaboration between researchers and practitioners was limited.

Additionally, both Danish EIAs did not indicate a collaboration with health professionals. This is represented within the EIAs by the coverage of mainly environmental factors. The EIA by DSB on the maintenance workshop only mentioned health impacts in correlation with harmful waste. The EIA of the biogas plant in the municipality of Svendborg concentrated on safety issues but also on potential positive health impacts due to a decrease in bacteria.

Within the EIA/SEAs, limited collaboration among environmental and health professionals led to fewer considerations of health impacts. Possible reasons for the constrained cooperation could be the lack of information regarding contacts to health professionals. Most practitioners of the survey worked in environmental-related sectors, which could also explain the lack of experience with health and health-related competencies within EIA/SEAs. Further, researchers or health consultants might not be aware of specific EIA/SEA procedures and methods, which could inhibit collaborations and prevent the assessment of relevant factors. This issue could be countered by educating health professionals regarding the procedures and methods of EIA/SEA. Collaboration between the health and environmental sectors could minimise the potential lack of experiences, which were emphasised as a main barrier for the inclusion of health into EIA/SEA by the respondents of the survey.

By reaching out directly to researchers within the field of public health, the aim was to assess their experiences regarding consultation on health issues within EIA/SEAs. Most of the ten researchers considered themselves HIA researchers. However, many researchers who conducted the survey had never been consulted in the process of conducting EIA/SEA. Their area of expertise, although in the area of health, might be too theoretical to transfer to the practical nature of EIA/SEAs. Hence, the researchers who completed the survey might not be equipped to consult within EIA/SEA. To achieve a realistic view on the topic of whether health professionals consult within EIA/SEAs, the question should be directed to health professionals with more practical work experience.

However, the lack of cooperation could also indicate an employment gap, which could be used as an opportunity by public health professionals to engage and impart health-related knowledge. This would be in line with many respondents’ emphasising that a public health background should be required to assess health within EIA/SEA. As the WHO suggested already in 2014, the cooperation of health and environmental sectors should be established as early as possible in the process of conducting an impact assessment [25]. The establishment of entry points for health professionals into the process of conducting EIA/SEA could foster cooperation agreements.

### 4.3. Guidance Documents

Guidelines are a basic tool used that can be used to implement environmental assessments and thereby facilitate comprehensive outcomes. Within the survey, respondents were asked in various ways on their current use and need of further guidance documents.

The respondents clearly emphasised the need for more guidance documents, and the lack of guidance as a barrier to include health into environmental assessments. This pertained to the need for more guidance regarding the inclusion of health, specific to the four types of assessment (plans, programs, projects and policies) and specific to the areas. More guidance documents specific to the types could clarify the starting point of the assessments, explain the meaning and implications for each type and elaborate whether there are issues to be addressed, activities to be implemented or work to be done in a specific timeframe. Further, the responsibility for preparing the report should be clarified. An example is Denmark, where EIAs on projects are the responsibility of the project proponent, often conducted by a consultant company, and EIAs of plans and programmes are conducted by an authority such as a municipality or ministry [14].

Not only did the respondents express the need for guidance documents generally and specific to the type, but also specific to the area, regarding the inclusion of health into EIA/SEA. In the survey, ten areas were listed as retrieved from Annex I and II of the 2011 EIA Directive and paragraph 10 and 11 of the SEA Directive [5,12]. In contrast to practitioners, fewer researchers expressed the need for more area specific guidance documents. A reason could be that researchers perceived the ten areas as a massive task. Although the development of ten separate guidance documents would be a considerable undertaking, it is not unmanageable. Many sections would be similar, such as the general content, general procedures or the development of the report. The guidance should, however, be tailored to the context of the respective area regarding the explanation of terms, scoping, appraisal parts with methods and analyses that can be employed within the specific area. This could enhance the assessments, as guidance could be more specific and tangible for practitioners, thereby presenting the opportunity for more comprehensive assessments of health factors.

The question is, however, who can develop these guidance documents? They should be developed by a reliable entity, and a further asset would be to have mutually agreed guidance documents within the EU. A reliable entity could be the WHO, or another established internationally operating organisation. This could ensure that the development of the documents would be both comprehensive and evidence based. While developing the documents, parties from all areas such as environment, health, economy and policy should be considered to establish a document that is tailored to the needs of all stakeholders involved and that is in line with the EC directives. To have mutually agreed guidance documents on EU level could result in a standardisation and simplification of conducting EIA/SEAs. The adaption to the national context is crucial to ensure the guidance document is readily applicable.

In the survey, almost all participants stated the use of guidance documents as a source when conducting EIA/SEA. Guidance documents could aid in suggesting suitable sources for the different stages of the assessment. It could specify on sources that could be used for assessing, examining and analysing quantitative as well as qualitative data. More specifically this could mean guidance for accessing scientific databases and literature. By that, procedures could be fostered, and the operation of the assessment could be enhanced.

The EC has produced guidance documents to assist practitioners in conducting EIAs while complying with the directives. However, an examination of the documents proved that many topics were only covered superficially. Especially the terms population and human health received little attention. Regarding the term *human health*, the document only provides a footnote stating that human health is a broad term and should be assessed on a context-specific basis [26].

All in all, guidance documents are crucial for the implementation of any impact assessments. Guidance documents should define terms and provide options for methods and analytical approaches to aid the assessment. National as well as international political support can initiate the process of developing guidance documents, it can foster cooperation between sectors and can result in accessible and transparent outputs.

### 4.4. Political Support

There are political incentives that can facilitate the use of HIAs or health assessments. Montenegro and Serbia are in the process of accession to the EU. Serbia officially applied for an EU membership in 2009 and Montenegro in 2008 [27]. The SEA of the Tivat airport extension in Montenegro and the SEA on the waste management plan in Serbia were conducted to comply with European standards, thus, making political support a facilitator in the implementation of the SEAs [16,18].

In the context of including population and human health into EIAs, Denmark showed limited political support. The VVM law does not elaborate on the inclusion of population and human health into EIA, and neither are the terms defined. This indicates a lack of political support on the assessment of health, which could in turn explain the low requirements for health within EIAs.

Some Danish practitioners who conducted the survey expressed the need for relevant laws pertaining to health within EIA. As the VVM law covers this area, the need could be translated to develop additions to the law, such as definitions of the terms related to health or enhanced examples regarding what health factors to consider. Definitions and examples could also be a part of guidance documents, which is currently not the case.

## 5. Conclusions

EIA and SEA are established and comprehensive tools to assess the significant effects of plans, programmes and projects on the environment. Since the amendment of the EIA Directive, effects on population and human health must be assessed to ensure the prevention of any adverse effects on health. The HIA is a comprehensive and established tool to assess health. Its approaches and methods to assess and analyse health impacts can aid EIA/SEAs. Due to the HIAs aim to assess all potential health impacts, the tool provides a natural basis for assessing direct and indirect as well as positive and negative health impacts.

The reviewed environmental assessments revealed that health determinants assessed within EIA/SEAs mostly pertained to environmental factors. Hence, there is a need for health assessments that go beyond environmental factors and include other relevant health determinants, such as the ones listed in Table 2. Nevertheless, the literature review showed that within the practice of conducting EIA/SEAs, there was an awareness to consider and assess health impacts.

The health assessment could be fostered by intersectoral cooperation between the environmental and health sectors within EIA/SEAs. As the WHO suggested already in 2014, the cooperation of health and environmental sectors should be established as early as possible in the process of conducting an impact assessment. Encouraging more health-related training and establishing potential entry points for health professionals into the process of conducting EIA/SEA could facilitate cooperation. Here, political support could promote practices to include health into EIA/SEA.

The comparison of researchers’ and practitioners’ views on the inclusion of health into environmental assessments aided in revealing their opinions. Due to the rather small sample size, their perceptions cannot represent the view of all researchers and practitioners working in the field. Establishing a survey based on a larger population could reveal crucial and representative opinions on the topic of health within EIA/SEAs. Nevertheless, the perceptions of researchers and practitioners regarding health inclusion into EIA/SEA showed that there is a need for more guidance documents. These guidance documents should elaborate on essential terms such as population and human health. Their meaning is crucial to develop comprehensive health assessments. The guidance documents can be established by international organisations such as the WHO as they represent reliable institutions. However, there must be an effort on national level as well, to translate those documents into the national language and contexts. This can ensure comprehensibility for those practically applying the guidance documents on environmental assessments.

To summarise this study, a list of recommendations is provided, resulting from the reviewed literature and the compared perceptions of practitioners and researchers to include health into EIA/SEA. The recommendations aim to inform decision making and official guidelines to ensure the consideration of the broader health determinants within environmental assessments. The list encompasses the most important and strongly emphasised points backed by the literature review and survey results presented throughout this study:Additional or enhanced guidance documents,Engagement of health professionals in the environmental sector,Health-related training, andPolitical support.

To generalise results on a European level, more research must be conducted. This could include a more comprehensive analysis of EIA/SEA reports within different countries. Moreover, a survey conducted in EU member states could generate information about health in environmental assessments as perceived by practitioners and researchers. However, the results of this article can be used to understand the complexity of integrating health into environmental assessments. Further, there might be similar aspects for integrating health into environmental assessments in other countries, which could be used to support decision making.

## Figures and Tables

**Table 1 ijerph-16-04570-t001:** Inclusion and exclusion criteria for scientific literature search.

Inclusion Criteria	Exclusion Criteria
Mentioning EIA, SEA, HIA	Only mentioning risk assessment
About policy, plan, programme, project, proposal and EIA, SEA, HIA	Only focusing on EIA, SEA, HIA appraisal/development of approach instead of implementation
About institutionalisation, integration or implementation of EIA, SEA, HIA	Only mentioning SIA, SA, HRIA, economic assessment or LCA ^1^
EIA, SEA mentioning HIA or health assessment	Only focusing on outcome or process evaluation

^1^ SIA: Social Impact Assessment; SA: Sustainability Assessment; HRIA: Human Right Impact Assessment; LCA: Life Cycle Assessment; SEA: Strategic Environmental Assessment; HIA: Health Impact Assessment; EIA: Environmental Impact Assessment.

**Table 2 ijerph-16-04570-t002:** Determinants of health as scope for EIA/SEA. Adapted from Nowacki, 2018 [7].

Scope
General social, economic and political factors
Environmental factors
Built environment and housing
Health services
Other public services and local economy
Private services and local economy factors
Employment and livelihood factors
Family and community structure
Behavioural risk factors
Biological factors

**Table 3 ijerph-16-04570-t003:** Comparison of reviewed literature.

Author	Year	Country	Study Design	Population	Intervention	Outcome
Josimović et al. [16]	2016	Montenegro	Case study	Tivat Municipality: about 14,000	SEA	**Health determinants:** Environmental factors; built environment and housing; health services; other public services
						**Considerations of vulnerable groups:** Yes ^1^
						**Incentive:** Legislation
Carvalho et al. [17]	2017	Lithuania/Latvia	Case study	N/A	SEA	**Health determinants:** Environmental factors; built environment; employment and livelihood
						**Considerations of vulnerable groups:** No
						**Incentive:** EU accession
Josimović et al. [18]	2015	Serbia	Case study	City of Belgrade: 1,621,396	SEA	**Health determinants**: Environmental factors; built environment; employment and livelihood
						**Considerations of vulnerable groups:** No
						**Incentive:** EU accession
COWI [19]	2019	Denmark	Case study/report	N/A	EIA	**Health determinants**: Environmental factors
						**Considerations of vulnerable groups:** No
						**Incentive:** Legislation
Dansk Biogasrådgivning A/S [20]	2019	Denmark	Case study/report	N/A	EIA	**Health determinants**: Environmental factors; biological factors
						**Considerations of vulnerable groups:** No
						**Incentive:** Legislation

^1^ Mentioned vulnerable facilities, not groups: “Health facilities, recreational facilities, housing, educational facilities” [16].

**Table 4 ijerph-16-04570-t004:** Characteristics of participants.

Number of all participants (n)	42
Number of EU practitioners (n)	7
Number of Danish practitioners (n)	25
Number of researchers (n)	10
Average age	46
Age range	29–64
Workplace	Europe

**Table 5 ijerph-16-04570-t005:** The table lists the three main barriers and facilitators to include health into EIA/SEA for practitioners and researchers, as well as sources used to conduct EIA/SEA by practitioners and researchers. The percentages (%) are calculated using the frequency of the answer (n) and dividing it through the total number of cases (N).

		% of All Practitioners (*N* = 32)	% of All Researchers (*N* = 10)
**Barriers**	Lack of experiences/competencies within the field	50% (*n =* 16)	50% (*n =* 5)
	Lack of guidance documents	44% (*n =* 14)	/
	Lack of awareness	41% (*n =* 13)	/
	Institutional barriers between different sectors	/	60% (*n =* 6)
	Lack of training	/	50% (*n =* 5)
**Facilitators**	Guidance documents	53% (*n =* 17)	50% (*n =* 5)
	Collaboration between different sectors	41% (*n =* 13)	80% (*n =* 8)
	Political support	41% (*n =* 13)	50% (*n =* 5)
**Sources for conducting EIA/SEA**	Guidance documents	94% (*n =* 30)	70% (*n =* 7)
	Scientific literature	78% (*n =* 25)	80% (*n =* 8)
	Expert opinion	78% (*n =* 25)	80% (*n =* 8)
	Grey literature	66% (*n =* 21)	70% (*n =* 7)
	Policy documents	63% (*n =* 20)	70% (*n =* 7)
	Statistical databases	56% (*n =* 18)	80% (*n =* 8)

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
