# Peer review of "Analysis of Health in Environmental Assessments—A Literature Review and Survey with a Focus on Denmark"

_ijerph, 2019, doi:10.3390/ijerph16224570_

Round 1
Reviewer 1 Report
Dear authors, please find below some comments and suggestions that will improve the transparency and robustness of your manuscript.
Methods-review. Please extend the methods here, language restrictions, search strategy, study selection, data extraction, and quality assessment. Please add supplementary material, the full search strategy, and the flow chart of the study selection process. Methods-survey. Please specify which countries from the EU were targeted, the extension of the survey, the number of questions, when was applied, how the researchers and practitioners were contacted, and recruitment techniques. Please also include the full survey in the supplemental material. Methods-comparative analysis. It is mentioned that was performed a new assessment in small groups, is this a type of discussion groups? Or the groups are analysis groups of the data resulted from the survey? Please clarify and add a more in-depth explanation of how the groups where chosen, how many groups, and why. Results-3.1. Please link this with the flow chart. Results- 3.1.1. Do you mean European studies? If so, please update. Results. Please add a table with the description of the papers included in the review — author, year, country, type of study, population, intervention, outcome, etc. Table 2. You mention the number of practitioners is 7, and then a number of Danish practitioners are 25. How does this work? One of those title or numbers are wrong. Do you mean a number of EU practitioners are seven or Danish participants is 25? Please specify or correct it. Table3,4, and 5. Please fusion these three tables. Discussion. I feel that the barriers need more recommendations in the discussion — for example, lack of training, and no to many mentions to training.Author Response
Please see the attachment

Reviewer 2 Report
General review of the article: There is a need for a more comprehensive health assessment within EIA and SEA to comply with the EIA and SEA Directives. Therefore, it is of great importance to establish the status of inclusion of assessment of impacts on health into EIA and SEA. However, the present research needs to be further well reorganized. Overall, I recommend publication after major revision and I provide following comments. Hope this helps.
Abstract. Page 1, Line 14, Please explain what is “o date”.
2.Page 3, Table 1., there is a spelling mistake about “focussing”.
3.In the second part, there is a section of 2.3 comparative methodology, however, there is no corresponding conclusion in the third part. In my opinion, you just classified the research objects, rather than a comparative study.
Page 4, 3.1 Literature Search. It is not comprehensive. There are only 4 articles in 3.1.1 and 2 articles in 3.1.2, and all of them are just narrations, lacking their own summary and analysis. Is this enough to grasp the current research trends? The review didn’t provide sufficient background, especially for the latest researches. Here are some key articles in the literature of health assessment into environmental impact assessment. This would allow contextualizing the paper in the relevant literatures.
https://www.ncbi.nlm.nih.gov/pmc/articles/PMC5615548/
https://link.springer.com/article/10.1007/s11771-018-3852-2
https://www.nature.com/articles/s41598-018-36511-z
In the part of 3.1. Pure text description with no graphics. It looks like a reading notes. Authors could put the results into tables or figures just like part 3.2 to let the readers read more vividly.
6.Page 5. In the part of 3.2 Survey. Line 194, is that enough to take the 42 respondents as you mentioned in order to represent occupations including international health organizations, regional or local health and environmental authorities, public health or environmental institutes, academic units and Non-governmental organizations ?.
Page 5, Line 207. In the sentence “only 14 of all practitioners and 10 of Danish practitioners…”, Please explain whether the preceding 14 contains the following 10, if so, this expression is inappropriate.
Page 5, Line 217. In the sentence “50% of all practitioners, 44% of the Danish…”, Please explain whether the preceding 50% contains the following44%, if so, this expression is inappropriate.
Page 6-8, from Table 3 to Table 5, what does the data in the tables mean? And how to calculate and then get it?
The discussion section deals with the main problems of this paper. It is very detailed.
There are many previous studies regarding the health in Environmental Assessments. How is the present study significant from other previous studies?
Reviewer 3 Report
This is a manuscript with mixed usage of literature review and a qualitative/quantitative survey, to study the effectiveness and necessity of including impacts on health in EIA/SEA. The intention of this manuscript is not very clear, whether it being for supporting regulatory decision making, or just for academic research interests? Also the overall approach of this study is focusing on Denmark only, without comparisons to countries outside of the EU, therefore the scope is limited. Anyway, I would like to point out some major flaws and have authors addressing those issues.
The title is too abstract and most readers can’t tell what is in the article, suggesting to have a little bit more details in the title, something like “a comprehensive review and survey of integrating HIA in EIA/SEA”. Otherwise the title is just being too ambiguous, What is the date range of the literature, it seems it is until 2019, but when does it start. Also how many of these are in the EU or Denmark? For the survey, can you attach these questions in the appendix or supporting materials? Without seeing these questions, I don’t know if they have biases or not. What do you mean by “go beyond environmental factors”, do you mean most EIA/SEA only focused on health outcomes directly affected by the environment? This is confusing. The “conclusion” of the study is reasonable, but not adequately supported by the study design, or at least based on the data available.Author Response
Please see the attachment

Round 2
Reviewer 1 Report
Thanks to the authors for providing satisfactory answers to my comments and suggestions.
Author Response
Dear reviewer,
thank you very much for your valuable comments in the peer-review process!
Reviewer 2 Report
The authors give detailed explanations (as well as corresponding corrections) to every of my question arisen during the first revision process. However, I’m not that satisfied with the new version of the manuscript. As the authors also have described, the sample basis of 42 respondents is rather small, the sample size of researchers was smaller. Hence, results were treated as the views of a non-representative group. This should be clarified in the conclusions, because 42 samples are definitely not enough to be used to support decision-making.
Author Response
Dear reviewer,
thank you for your valuable inputs to our manuscript within the peer-review process. To address the sample size issue also in the Conclusion, we added the following sentence (5. Conclusion, from line 533):
"The comparison of researchers’ and practitioners’ views on the inclusion of health into environmental assessments aided in revealing their opinions. Due to the rather small sample size, their perceptions cannot represent the view of all researchers and practitioners working in the field. Establishing a survey based on a larger population could reveal crucial and representative opinions on the topic of health within EIA/SEAs. Nevertheless, the perceptions of researchers’ and practitioners’ regarding health inclusion into EIA/SEA showed that there is a […]."
We hope the added paragraph addresses the issue of the sample size appropriately in the conclusion.
Thank you very much!